# Contextual Biasing with the Knuth-Morris-Pratt Matching Algorithm

## Abstract

Contextual biasing refers to the problem of biasing the automatic speech recognition (ASR) systems towards rare entities that are relevant to the specific user or application scenarios. We propose algorithms for contextual biasing based on the Knuth-Morris-Pratt algorithm for pattern matching. During beam search, we boost the score of a token extension if it extends matching into a set of biasing phrases. Our method simulates the classical approaches often implemented in the weighted finite state transducer (WFST) framework, but avoids the FST language altogether, with careful considerations on memory footprint and efficiency on tensor processing units (TPUs) by vectorization. Without introducing additional model parameters, our method achieves significant word error rate (WER) reductions on biasing test sets by itself, and yields further performance gain when combined with a model-based biasing method.

## 1 Introduction

Recent years have seen a tremendous explosion in voice user interfaces (VUIs), like voice search, assistant, and control applications. The success of VUI-based applications depends on the ability of the underlying Automatic Speech Recognition (ASR) system to properly transcribe phrases that are contextually relevant to the speaker, the listener, or both. Examples of contextually-relevant phrases include names of the speaker's contacts and geographically-close points of interest. Contextually-relevant phrases are inherently hard to recognize because they represent instances of domain shift. For example, generally, it is much more likely for a single user to speak the name of one of their contacts than for any given contact name to occur in a given training data set; indeed, a given name or phrase may not appear at all in an ASR system's training set in the case of unorthodox spellings (Ke\$ha) or novel words (COVID-19). Further, contextually-relevant phrases may not be known until inference time, e.g., as the user of a voice assistant can add contact names any time before speaking.

ASR *contextual biasing* is a set of techniques which enables ASR systems to recognize contextually-relevant words without retraining. Contextual biasing can generally be grouped into model-based and inference-based approaches. Model-based methods typically incorporate a biasing component into an end-to-end (E2E) ASR system (Graves, 2012; Chorowski et al., 2015; Chan et al., 2016a), which takes in biasing contexts as additional input to the E2E model. An attention mechanism (Vaswani et al., 2017) is typically used to condition the model outputs on biasing contexts (Munkhdalai et al., 2021; Chang et al., 2021; Han et al., 2022) (see Sec 3 for more discussions).

The more classical inference-based approach, dating back to the pre-E2E era, injects biasing contexts to boost decoding scores for the words or phrases in the biasing contexts to increase the probability of recognizing those words (Aleksic et al., 2015; Hall et al., 2015). A compact search graph, based on Weighted Finite State Transducers (WFSTs, Mohri et al., 2002), is built to encompass the set of biasing phrases, and incorporated into the normal search graph which then transduces acoustic model outputs to word-level hypotheses. Weights are distributed along edges of the biasing search graph, so that when the acoustic model output extends the matching of the phrases, a bonus score is added to the hypothesis to help it survive beam search and increase its likelihood of becoming the top hypothesis. The approach was later extended to E2E models (Zhao et al., 2019) where bonuses are incorporated at subword level. While E2E ASR systems have greatly simplified modeling and deployment, and that most components are readily implemented on GPU or TPU to enjoy parallel processing, FST-based biasing poses significant challenges for an efficient TPU-based implementation, due to their inherent sparse nature (adjacency matrices for FSTs are typically very sparse).

**Our contributions**   In this work, we propose a TPU-friendly implementation of search-based biasing, leveraging the equivalence between the biasing FST and the efficient matching algorithm by Knuth et al. (1977), with careful considerations on memory complexity and efficiency through vectorization. Our algorithms can be incorporated into the beam search of any ASR system, in both the on-the-fly rescoring and shallow fusion manner. On large voice search datasets, our method achieves significant word error rate (WER) reductions on biasing test sets by itself, without introducing additional model parameters. And when plugged into a model-based biasing method, namely neural associative memory (NAM, Munkhdalai et al., 2021), our method leads to further improved biasing accuracy. Our method enables learning with the discrete structure of ASR biasing, and can be potentially useful for other sequence transduction tasks.

## 2   OUR METHOD

An intuitive and classical idea for biasing is to check iteratively, at each beam search step, if the suffixes of partial hypotheses are partially or fully matching any of the biasing phrases, and give score bonuses to those with matches. This helps a partial hypotheses to survive beam search pruning, if it has the potential to develop into a full match of a biasing phrase. In this section, we develop the algorithms for efficiently performing pattern matching for multiple biasing phrases, and properly assigning biasing bonus for each beam search expansion, based on the classical KMP algorithm for string/pattern matching. We review the classical algorithm in Sec 2.1, describe its usage for biasing in Sec 2.2, discuss the two variants for beam search in Sec 2.3, and an extension in Sec 2.4.

**Notations**   Below we use $\mathcal{P}$ to denote the pattern sequence to be searched/matched, and $\mathcal{T}$ to denote the sequence to be searched from; both are strings in the context of the classical matching algorithm or token sequences in the context of speech recognition. The length of $\mathcal{P}$ is denoted $len(\mathcal{P})$. We use $\mathcal{P}[i]$ to denote the element at (0-based) index $i$, and use $\mathcal{P}[s,\ldots,t] := [\mathcal{P}[s],\mathcal{P}[s+1],\ldots,\mathcal{P}[t]]$ to denote the sub-sequence of $\mathcal{P}$ with start index $s$ and end index $t$. Two sequences are equal if they have the same length and corresponding elements match for all indices.

### 2.1   THE KNUTH-MORRIS-PRATT MATCHING ALGORITHM

For searching the occurrences of a string $\mathcal{P}$ of length $m$ within another string $\mathcal{T}$ of length $n$, the most naive solution is perhaps to loop over the set of indices $j = 0, 1, \ldots, n - m$, and check if the sub-string $\mathcal{T}[j,\ldots,j+m-1] = \mathcal{P}$, which requires another loop over the elements of $\mathcal{P}$. Clearly, this algorithm has a worse-case time complexity of $\mathcal{O}(mn)$. There exists, however, a much more efficient linear-time Knuth-Morris-Pratt (KMP) matching algorithm (Knuth et al., 1977) for this problem, with a worse case complexity of $\mathcal{O}(m + n)$. We extract two major components out of KMP below, which are used for efficiently maintaining status of matching, as needed by biasing.

#### 2.1.1   THE FAILURE FUNCTION

The key insight behind the KMP algorithm is to not waste comparisons: if during matching we have a partial matching of length $i$ and $\mathcal{T}[j] \neq \mathcal{P}[i]$, then instead of moving back to index $j - i + 1$ for $\mathcal{T}$ and moving back to index 0 for $\mathcal{P}$, and restarting the matching (by checking whether $\mathcal{T}[j - i + 1, \ldots, j - i + m] = \mathcal{P}$), we may continue by comparing $\mathcal{T}[j]$ against $\mathcal{P}[\pi(i)]$ with some $\pi(i) < i$, without backtracking in $\mathcal{T}$. Here $\pi(i)$ specifies the index of the *potential* next match in $\mathcal{P}$ when we have a mismatch for $\mathcal{P}[i]$, and is called the *failure function*.

The failure function is originally defined as follows (Cormen et al., 2001): set $\pi(0) = -1$, and for $i = 1, \ldots, m - 1$,

$$\pi(i) = \max \left\{ k < i : \ \mathcal{P}[0, \ldots, k-1] = \mathcal{P}[i-k, \ldots, i-1] \right\}.$$

That is, for $i > 0$, $\pi(i)$ is the length of the longest proper prefix that matches a proper suffix of the sequence $\mathcal{P}[0, \ldots, i-1]$; the value is 0 if no such prefix exists. The special value $-1$ indicates that there is no possible match starting at the current index of $\mathcal{T}$ and we must move to the next index to restart matching: if $\mathcal{T}[j] \neq \mathcal{P}[0]$, we must move to index $j + 1$ in $\mathcal{T}$ to compare again with $\mathcal{P}[0]$.

To see why this definition helps save unnecessary comparisons, consider the scenario where we have a partial match of length $i > 0$, but then the mismatch $\mathcal{T}[j] \neq \mathcal{P}[i]$ occurs. Since $\mathcal{T}[j-i,\ldots,j-1] = \mathcal{P}[0,\ldots,i-1]$, we must have

$$\mathcal{T}[j - \pi(i), \ldots, j - 1] = \mathcal{P}[i - \pi(i), \ldots, i - 1] = \mathcal{P}[0, \ldots, \pi(i) - 1].$$

---

**Algorithm 1** Forward a search pattern with an input token.

---

**Input:** $\mathcal{P}$ with length $m$ and failure function $\Pi$, current partial matching length $i$, new token $x$.
**Output:** Updated partial matching length $q$, and if we obtain a full match, after consuming $x$.

   **procedure** FORWARD($\mathcal{P}, i, x$)
      $full\_match \leftarrow False$
      **if** $\mathcal{P}[i] = x$ **then**
         $q \leftarrow i + 1$
         **if** $q = m$ **then**                                   ▷ Full match
            $full\_match \leftarrow True, \quad q \leftarrow 0$
         **end if**
      **else**
         $k \leftarrow \Pi[i]$
         **while** $k >= 0$ and $\mathcal{P}[k] \neq x$ **do**               ▷ Determinization loop
            $k \leftarrow \Pi[k]$
         **end while**
         $q \leftarrow k + 1$                         ▷ Either $k = -1$ or $\mathcal{P}[k] = x$
      **end if**
      **return** ($q, full\_match$)
   **end procedure**

---

Therefore, without backtracking in $\mathcal{T}$, we already have a partial match of length $\pi(i) < i$, and we then check if $\mathcal{T}[j] = \mathcal{P}[\pi(i)]$ to determine whether we can extend the partial match; in case of further mismatch, we repeat the process and backtrack to $\pi(\pi(i))$, $\pi^3(i)$,..., etc, until we reach $-1$.

The failure function we use in this work, denoted as $\bar{\pi}(\cdot)$, is based on the above definition, and has an additional "shortcut" logic (Aho & Corasick, 1975): for $i = 1, \ldots, m - 1$,

$$\bar{\pi}(i) = \begin{cases} \bar{\pi}(\pi(i)) & \text{if } \mathcal{P}[\pi(i)] = \mathcal{P}[i], \quad \text{(shortcut)} \\ \pi(i) & \text{otherwise.} \end{cases}$$

The rationale behind the shortcut is that, as we are backtracking due to $\mathcal{T}[j] \neq \mathcal{P}[i]$, in the case of $\mathcal{P}[\pi(i)] = \mathcal{P}[i]$ we deduce $\mathcal{T}[j] \neq \mathcal{P}[\pi(i)]$, and thus $\pi(i)$ cannot be the next possible match and we shall keep backtracking. We provide the algorithm for computing $\bar{\pi}(\cdot)$ in Algorithm 4 (Append A). The time complexity for building the failure function of a pattern with length $m$ is $\mathcal{O}(m)$.

An example of search pattern and its failure function is as follows.

| $i$ | 0 | 1 | 2 | 3 | 4 | 5 | 6 | 7 | 8 |
|---|---|---|---|---|---|---|---|---|---|
| $\mathcal{P}[i]$ | A | B | A | C | A | B | A | B | A |
| $\Pi[i]$ | -1 | 0 | -1 | 1 | -1 | 0 | -1 | 3 | -1 |

(1)

### 2.1.2 THE FORWARD FUNCTION

With the failure function defined above, we can define a forward function. Given the matching state, defined as the current partial matching length $i$ (i.e., we have matched $\mathcal{P}[0, \ldots, i - 1]$ so far, and $i$ is the next index to match in $\mathcal{P}$), and a new token $x$ from the string $\mathcal{T}$ to be searched, the forward returns the updated partial matching length (the new position in $\mathcal{P}$ to match), after *consuming* $x$. Here by "consuming" we mean either we have a match for $x$ and we move to $i + 1$ in $\mathcal{P}$, or we determine that it is impossible to match $x$ and restart the matching; in both cases we move beyond $x$ in $\mathcal{T}$. The logic is sketched in Algorithm 1.

The complexity of this algorithm mainly lies in the "determinization loop", where we keep backtracking until we find a match of $x$ in $\mathcal{P}$; when no such match is possible, we end up with $k = -1$ out of the loop, and restart matching at the next token in $\mathcal{T}$. Additionally, we check whether we obtain a full match of $\mathcal{P}$ after matching token $x$, in which case we also restart matching at the next token in $\mathcal{T}$ (we are not interested in overlapping matches of patterns in this work).

If we add another loop on top of Algorithm 1 over the tokens in $\mathcal{T}$, we recover the KMP search algorithm, which has a time complexity of $\mathcal{O}(n)$ after the failure function is computed (with $\mathcal{O}(m)$ complexity). Note how similar is the determinization loop of Algorithm 1 to the inner loop of Algorithm 4; in fact the latter can be seen as searching $\mathcal{P}$ over itself.

---

**Algorithm 2** Compute bonus score of a token extension.

---

**Input:** Biasing phrases $\{\mathcal{P}^b\}_{b=1}^B$, current partial matching lengths $\mathcal{I} = (i^1, \dots, i^B)$, new token $x$.
**Output:** Updated partial matching lengths, and biasing bonus.
   **procedure** COMPUTEBONUS($\{\mathcal{P}^b\}_{b=1}^B, \mathcal{I}, x$)
      $any\_match \leftarrow False$                       ▷ Track if there is any full match
      **for** $b = 1, \dots, B$ **do**
         $u^b, \ match^b \leftarrow$ FORWARD($\mathcal{P}^b, i^b, x$)
         **if** $match^b$ **then**
            $any\_match \leftarrow True$
            $v^b \leftarrow len(\mathcal{P}^b)$       ▷ For full match, use pattern length for computing potential
         **else**
            $v^b \leftarrow u^b$
         **end if**
      **end for**
      $bonus \leftarrow \mu(v^1, \dots, v^B) - \mu(i^1, \dots, i^B)$
      **if** $any\_match$ **then**
         $u^b \leftarrow 0$   for   $b = 1, \dots, B$     ▷ In case of any full match, restart matching for all phrases
      **end if**
      **return** $((u^1, \dots, u^B), \ bonus)$
   **end procedure**

---

We can design a finite state automaton (FSA) $\mathcal{A}(\mathcal{P})$ with $m$ states, where state $i = 0, \dots, m - 1$ denotes the state for partially matching $i$ tokens of $\mathcal{P}$, and the forward function provides the transition function for this automaton, i.e., for an arc that starts at state $i$ with input $x$, it ends at the state specified by FORWARD($\mathcal{P}, i, x$). With the determinization loop, each transition consumes a non-epsilon token on its edge, ensuring that $\mathcal{A}(\mathcal{P})$ is deterministic and epsilon-free. See Cormen et al. (2001, Chapter 32.4) for more detailed discussions on the equivalence between KMP and FSA.

One could run Algorithm 1 for all $x$ in the vocabulary (all characters in the alphabet in the case of string matching) for $i = 0, \dots, m - 1$; this yields a table of size $m \times |V|$ where $|V|$ is the vocabulary size. While we could in principle use this table for biasing, the memory cost may be too high when we have on the level of thousands or more patterns to search, each with some number of tokens (up to 16 in our experiments), while $V$ is also in the thousands (4096 for our ASR system). It is therefore much more memory efficient to store the failure function which only takes $\mathcal{O}(m)$ memory, and we pay the cost of determinization loop. For any $x$, the number of times we have to backtrack in the determinization loop is bounded by

$$\gamma(\mathcal{P}) = \max_i \ e_i, \quad \text{where } e_i \text{ is the integer satisfying } \pi^{e_i}(i) = -1. \tag{2}$$

As an example, for the pattern in (1), we have $\gamma(\mathcal{P}) = 3$ with maximum achieved at $i = 7$.

## 2.2 CONTEXTUAL BIASING WITH KMP

For biasing in ASR, each utterance is associated with $B$ biasing phrases, denoted as $(\mathcal{P}^1, \dots, \mathcal{P}^B)$, and we attempt to match all of them at each beam search step. Another task is to assign a *bonus*, either positive or negative, to each new token expansion proposed by beam search. We achieve this goal by defining a *potential* function based on the state of matching.

For each phrase $\mathcal{P}^b$, $b = 1, \dots, B$, we first define a *scoring* function for partial matching of length $i$ (i.e., we have matched $\mathcal{P}^b[0, \dots, i - 1]$ so far). In this work, we simply parameterize the function to be linear in $i$:

$$f(\mathcal{P}^b, i) = i \cdot \delta, \qquad \text{for} \quad i = 0, \dots, len(\mathcal{P}^b),$$

where $\delta \geq 0$ is the per-token bonus and is a hyper-parameter to be tuned. It is future work to explore more sophisticated scoring functions for biasing phrases.

Let the partial matching lengths for the $B$ biasing phrases be $\mathcal{I} = (i^1, \dots, i^B)$. We define the potential function as the maximum scoring function over phrases:

$$\mu(i^1, \dots, i^B) = \max_{b=1,\dots,B} f(\mathcal{P}^b, i^b).$$

---

**Algorithm 3** Beam search with KMP biasing.

---

**Input:** ASR model. Biasing phrases $\{\mathcal{P}^b\}_{b=1}^B$. Beam size $K$. Number of biasing expansion $F$.
**Output:** Top $K$ hypotheses.
  **procedure** BEAMSEARCHWITHBIASING($\{\mathcal{P}^b\}_{b=1}^B, K, F$)
    $\mathcal{H} \leftarrow \{(h, s, \mathcal{I})\}$ where $h$ is empty with score $s = 0$, $\mathcal{I}$ contains biasing states of all zeros.
    **for** $step = 1, 2, \ldots,$ **do**
      $\mathcal{G} \leftarrow \{\}$                               ▷ $\mathcal{G}$ is buffer for storing intermediate hypotheses
      **for** $(h, s, \mathcal{I}) \in \mathcal{H}$ **do**
        Conditioned on $h$, compute top $F$ expansions $\{(x_1, s_1), \ldots, (x_F, s_F)\}$ with ASR model, where $x_k$ is token id and $s_k$ is its model score
        **for** $k = 1, \ldots F$ **do**
          $h' \leftarrow \text{Append}(h, x_k)$, $s' \leftarrow s + s_k$      ▷ Extend previous hypothesis by a token

> Option I (Shallow Fusion):
> $\mathcal{I}'$, $bonus \leftarrow \texttt{ComputeBonus}(\{\mathcal{P}^b\}_{b=1}^B, \mathcal{I}, x_k)$,   $s' \leftarrow s' + bonus$

          $\mathcal{G} \leftarrow \mathcal{G} \cup \{(h', s', \mathcal{I}')\}$
        **end for**
        $\mathcal{H} \leftarrow \text{Prune}(\mathcal{G}, K)$                ▷ Prune to top $K$ hypotheses
      **end for**

> Option II (On-the-fly Rescoring):
>   $\mathcal{G} \leftarrow \{\}$
>   **for** $(h, s, \mathcal{I}) \in \mathcal{H}$ **do**            ▷ There are exactly $K$ hypotheses in $\mathcal{H}$
>     $x \leftarrow h[-1]$                   ▷ Retrieve the last token
>     $\mathcal{I}'$, $bonus \leftarrow \texttt{ComputeBonus}(\{\mathcal{P}^b\}_{b=1}^B, \mathcal{I}, x)$,   $s' \leftarrow s + bonus$
>     $\mathcal{G} \leftarrow \mathcal{G} \cup \{(h, s', \mathcal{I}')\}$
>   **end for**
>   $\mathcal{H} \leftarrow \mathcal{G}$

    **end for**
    **return** $\mathcal{H}$

---

After consuming a new token $x$ for each biasing phrase with the forward function, the partial matching lengths are updated, based on which we compute the new potential function; the difference between the potentials is the bonus for $x$. We sketch this algorithm in Algorithm 2. We additionally track if we finish matching any phrase fully, in which case we restart matching for all phrases as we do not want overlapping matches. Note it is possible that $x$ extends matching for multiple phrases, especially if these phrases share prefix. If the hypothesis was partially matching a phrase and then becomes non-matching after consuming a new token, the previously added biasing bonus is canceled (Zhao et al., 2019).

To summarize, we maintain a total of $B$ integers as states for tracking the progress on each phrase. Consuming a new token and computing its bonus boils down to running the forward function. We vectorize the **for**-loop in Algorithm 2, and compute the forward functions for all $B$ phrases in parallel, which further reduces to looking up the failure function table and running the determinization loop in parallel. Therefore, the time complexity for Algorithm 2 is $\mathcal{O}(\bar{\gamma}B)$, where $\bar{\gamma} = \max_{b=1,\ldots,B} \gamma(\mathcal{P}^b)$ with the $\gamma$ function defined in (2). Note $\gamma$ is a worse-case bound for the number of iterations in the determinization loop.

## 2.3 INTEGRATING BIASING INTO BEAM SEARCH

We propose two ways to incorporate biasing bonus computation into beam search, with trade offs between accuracy and efficiency. We collectively refer to them as *KMP biasing*.

- **Shallow fusion**. In this approach, we perform biasing before pruning: for each hypothesis, we consider a number of top expansions according to the ASR model scores, and compute biasing bonus for each of them, which are combined with ASR scores used for pruning; this is similar to the shallow fusion approach for applying language models to ASR inference (Gulcehre et al., 2015; Chorowski & Jaitly, 2017; Zhao et al., 2019).

- **On-the-fly (OTF) rescoring**. In this approach, after expansions and pruning, we compute biasing bonuses for the expansion tokens of surviving hypotheses, and incorporate the bonus into the total score of each hypothesis, to influence future steps. Note this is different from offline rescoring, which only modifies total scores for re-ranking final hypotheses.

The two approaches are sketched in Algorithm 3. Let the beam size be $K$, which is the number of hypotheses maintained at the end of each beam search step. If we consider $F$ expansions for biasing, the complexity of shallow fusion is $\mathcal{O}(\bar{\gamma}KFB)$ per beam search step. Typically the biasing accuracy improves with $F$, at the cost of heavier computation. On the other hand, since we consider a total of $K$ expansions in on-the-fly biasing, its time complexity is $\mathcal{O}(\bar{\gamma}KB)$, cheaper than shallow fusion biasing by a factor of $F$. As we have multiple hypotheses, each of which considers multiple extensions, our implementation of `ComputeBonus` is parallelized in the (hyp, extension, phrase) combination dimension. The failure functions are computed for all phrases once before beam search starts. One can implement the loops using the `while` statement, and table lookups using `gather` or `einsum` functions provided by tensor-based learning platforms.

## 2.4 BOOSTING BIASING STRENGTH WITH PREFIXES

In many applications, the biasing phrase frequently follows a set of prefixes (also known as carrier phrases). For example, when using smart devices to initiate communication, the user typically speakers "call", "text", or "send a message to" before the contact name. It is natural to bias the ASR system more heavily towards the user's contact list, conditioned on recognizing such prefixes (Zhao et al., 2019). A naive way to extend our method to leverage prefixes is to augment the original biasing phrases (contact names in the above use case) with all combinations of prefix and biasing phrase ("call John", "text Joe", etc). If we have $C$ prefixes and $B$ biasing phrases, this approach leads to $B + CB$ phrases, significantly increasing the cost of KMP biasing.

We propose an alternative and more time efficient approach, with minimal cost increase in state maintenance. For each new token, we perform matching for both prefixes and biasing phrases simultaneously (although the hypothesis receives no bonus from matching prefixes), with a time complexity of $\mathcal{O}(C+B)$. If a new token is not extending the partial matching of any biasing phrase, but leads to a full matching of some prefix, we restart matching of biasing phrases for the extended hypothesis, which is marked as prefix-matching for all biasing phrases. And if a hypothesis is prefix-matching for some biasing phrase, we boost the scoring function of partial matches *of that biasing phrase* by a factor $\lambda > 1$. A hypothesis stays prefix-matching if it was prefix-matching, and the new token extends the partial matching of the same biasing phrase. Compared with the case without prefixes, we pay an additional cost of maintaining the states of partial matching lengths for prefixes, with a memory cost of $\mathcal{O}(C)$, and whether each hypothesis is prefix-matching for each biasing phrase, with a memory cost of $\mathcal{O}(B)$. We sketch the implementation in Algorithm 5 (Appendix B).

Our approach can be interpreted in the WFST framework, as having one FST for the set of prefixes and another for the set of biasing phrases, and we transit from the prefix FST to biasing FST when detecting full match of some prefix, so that the two FSTs are concatenated.

## 3 RELATED WORKS

**WFST-based biasing** Initial WFST (Mohri et al., 2002) approaches to contextual ASR (Aleksic et al., 2015; Hall et al., 2015) performed on-the-fly rescoring (Hori et al., 2007) during beam-search, for classical ASR systems that use a CLG decoder graph (Mohri et al., 2008). The contextual phrases are encoded separately from the CLG in a word-level deterministic WFST with failure transitions. Arbitrary word-level rescoring functions can be used, including CLG score replacement and various forms of interpolation. In Vasserman et al. (2016), the approach was extended to efficiently handle dynamic classes, by encoding non-terminal labels in the contextual models. Classes are dynamically inserted in the CLG graph, instead of being inserted in the contextual WFST, avoiding its exponential growth during determinization. Search errors caused by the late integration of contextual models at word labels were reduced by Williams & Aleksic (2017).

Later End-to-End (E2E) ASR systems most often do not have an external LM and require an alternative WFST approach that uses shallow fusion (Zhao et al., 2019) instead of on-the-fly rescoring. In this approach, the contextual information is encoded as a subword-level deterministic WFST with failure transitions, which is used to directly modify the acoustic scores, before pruning is done by

beam-search. The search space of E2E ASR systems tends to be sparser than the search space of classic ASR systems, so earlier integration is necessary to reduce search errors.

WFST contextual modeling can also be approached as a lattice-augmentation problem (Serrino et al., 2019; Huang et al., 2020). These techniques identify spans in the word lattice where rare entities are likely to occur and search for acoustically confusable alternatives that are contextually relevant. The span identification and fuzzy matching are done using flexible and efficient WFST-based techniques.

We note that FSTs are good at encoding domain knowledge and complex matching rules compactly. While they can be represented as graph with sparse adjacency matrices, in general FSTs are not efficient to use on TPUs which are optimized for dense operations. Our work is one step towards incorporating FST functionalities into a TPU-friendly implementation.

**Model-based biasing**  Context can be utilized by adding trainable parameters to the ASR model and performing *model-based biasing* (Fu et al., 2023; Harding et al., 2023; Xu et al., 2023). Learning such parameters in an end-to-end fashion was first considered in the CLAS model (Pundak et al., 2018), that augmented the LAS (Chan et al., 2016b) decoder with a suitable attention mechanism. CLAS is trained by sampling random n-grams (playing to role of bias phrases) from the reference transcripts. CLAS sampling was later improved in Alon et al. (2019) by emphasizing proper nouns, and considering hard phonetically-similar distractors (anti-bias terms). A notable drawback from CLAS is that the full-context nature of the decoder limits it to non-streaming applications.

The above limitation was addressed in Neural Associative Memory (NAM, Munkhdalai et al., 2021), a *streaming* model-based biasing method that utilizes an external associative memory module (Munkhdalai et al., 2019; Ramsauer et al., 2020) as an intermediate representation of biasing contexts, and is augmenting the RNN-T architecture. Given a trained ASR model, let $\mathbf{x}$ be audio feature sequence extracted by the encoder, and $\mathbf{y}$ be the label sequence. NAM learns a modified conditional probability $p(\mathbf{y}|\mathbf{x} + \Delta)$ by incorporating into ASR model an additional feature sequence $\Delta$. To compute $\Delta$, NAM utilizes an additional text encoder to extract embeddings of biasing phrases $\{\mathcal{P}^b\}_{b=1}^{B}$, which are used to construct the associative memory, and another Multi-Head Attention (Vaswani et al., 2017) module that uses $\mathbf{x}$ as query and the associative memory as keys and values, whose output context vector becomes $\Delta$. Essentially, the attention module is used to detect the presence of biasing phrases in the audio. NAM is trained as part of the E2E model (typically with the base ASR model frozen), so that the likelihood of ground truth, including the biasing phrase present in the audio, is maximized. At inference time, NAM introduces a biasing strength parameter $s \geq 0$ to control the effect of external biasing phrases (Wu et al., 2023), and uses $p(\mathbf{y}|\mathbf{x} + s \cdot \Delta)$ for decoding. Given that NAM injects biasing information at the encoder output, while KMP biasing works at beam search, they can be complimentary to each other, as is observed in our experiments.

## 4 EXPERIMENTS

We use a large RNN-Transducer (RNN-T, Graves, 2012) as our base ASR model. Our training set contains 520M utterances of English voice search queries; the total amount of audio is 490K hours. A small percentage of the training data is human transcribed while the rest is pseudo-labeled by a teacher (Hwang et al., 2022). We tokenize training transcripts, as well as biasing phrases, using a word-piece model (Schuster & Nakajima, 2012) with an inventory of 4096 tokens. All acoustic and text training data is anonymized and adheres to Google AI Principles (Google, 2023).

We use 128-dimensional log Mel-filterbank energies, extracted from 32ms window and 10ms shift, as frontend features. After two 2D-convolution layers, both with strides 2, the resulting feature sequence has a frame rate of 40ms and becomes the input to a conformer encoder (Gulati et al., 2020). The encoder consists of 16 Conformer layers of attention dimension 1536, where each attention module has 8 heads, and each feedforward network has a hidden dimension of 6144. The RNN-T decoder uses a $|V|^2$ embedding prediction network (Botros et al., 2021), which computes text features based on two previous non-blank tokens. The ASR model has a total of 870M parameters. For decoding, we perform label synchronous beam search with beam size $K = 8$. RNN-T has a special blank token which indicates non-emission and does not alter decoder and biasing states. The word error rate (WER) of our RNN-T system on an in-house test set of voice search queries is 3.8%.

**Evaluation**  We use both voice-assistant based real-audio data and TTS synthetic data as described in Munkhdalai et al. (2023) for evaluation. The real-audio test-set Contact-Tag contains 7.6K ut-

Table 1: WER (%) results obtained by KMP biasing.

| Dataset | KMP Biasing | | | |
|---|---|---|---|---|
| | OTF Rescoring $\delta = 2.4$ | Fusion $F = 10$ $\delta = 2.2$ | Fusion $F = 50$ $\delta = 2.3$ | Fusion $F = 4096$ $\delta = 2.3$ |
| Anti-Biasing, without-biasing WER: **1.7** | | | | |
| $B = 150$ | 1.7 | 1.7 | 1.7 | 1.7 |
| $B = 600$ | 1.8 | 1.8 | 1.8 | 1.8 |
| $B = 3000$ | 2.1 | 2.2 | 2.3 | 2.3 |
| With-Prefix, without-biasing WER: 9.6 | | | | |
| $B = 150$ | 4.1 | 3.7 | 2.6 | **2.4** |
| $B = 600$ | 4.5 | 4.0 | 2.9 | **2.7** |
| $B = 3000$ | 5.1 | 4.6 | 3.8 | **3.6** |
| Without-Prefix, without-biasing WER: 20.9 | | | | |
| $B = 150$ | 7.9 | 7.7 | 5.5 | **4.8** |
| $B = 600$ | 8.4 | 8.0 | 5.8 | **5.3** |
| $B = 3000$ | 10.1 | 9.6 | 7.9 | **7.4** |
| Contact-Tag, without-biasing WER: 14.7 | | | | |
| $B = 265$ | 8.7 | 8.3 | 7.8 | **7.7** |

terances focusing on contact recognition (i.e., call \$CONTACTS), each utterance is associated with 265 biasing entities and one of which is the true contact. The TTS data contains three categories: 1. **Anti-Biasing**: Utterances simulating general voice-assistant traffic (e.g., what's the weather), we use a super-set of that used in Munkhdalai et al. (2023) containing 10K utterances; 2. **With-Prefix**: 2.6K utterances with patterns such as: open \$APPS, call \$CONTACT, play \$SONG; 3. **Without-Prefix**: 1.3K Utterances with prefix-less patterns such as \$APPS, \$CONTACTS, or \$SONGS. The real-audio test set is anonymized and adheres to Google AI Principles (Google, 2023).

The utterances are associated with up to 3K biasing entities in total, and the maximum number of tokens in a biasing phrase is 16. With-Prefix and Without-Prefix evaluate in-domain performance (one of the biasing entities appears in the transcript truth), while Anti-Biasing evaluates out-of-domain performance (biasing entities are irrelevant to the transcript truth). In general, a larger set of biasing entities leads to more confusion in the ASR model and worse WER. We tune the hyper-parameters of methods based on averaged WERs on Anti-Biasing and With-Prefix; Without-Prefix and Contact-Tag are treated as test sets.

### 4.1 Results by KMP biasing

We first present the results of KMP biasing by itself, without combining with NAM or score boosting by prefixes. In both the OTF rescoring and shallow fusion modes, we tune the hyper-parameter $\delta$ which is the biasing bonus per token along phrases. We observe that, as we increase $\delta$, WERs on biasing sets first decrease, then stay low for a range of values, and eventually increase. We provide WER curves of varying $\delta$ in Figure 1 (Appendix C) for both modes.

We provide WERs together with the optimal $\delta$ in Table 1, for OTF-rescoring and three settings of $F$ for shallow fusion. Our method achieves significant WER reduction over the base ASR model on all biasing sets, e.g., by 50% to over 70% relative on the With-Prefix set with $B = 150$ phrases, while not degrading the Anti-Biasing set by much. We observe that shallow fusion consistently outperforms OTF rescoring as expected, and in general large $F$ leads to better WER. From $F = 50$ to the full vocabulary size 4096, improvements start to saturate and we find $F = 50$ to offer a good balance between accuracy and efficiency and use it in the experiments below.

### 4.2 Combining KMP biasing with model-based biasing

Given that KMP biasing is applied during beam search and is agnostic to the base ASR model, one may wonder if its performance gain is additive to that of a strong model-based biasing method. We train a state of the art NAM model on top of our base model, and its performance with normal beam search is provided in Table 2 (left column). We observe that model-based biasing does achieve superior results by itself, especially on Without-Prefix and Contact-Tag.

Table 2: WER (%) results obtained by NAM + KMP biasing. $s$ denotes NAM biasing strength, $\lambda$ denotes score boosting factor with prefixes.

| Dataset | NAM | NAM ($s = 0.5$) + KMP biasing | | NAM ($s = 0.5$) + KMP w. boost | |
|---|---|---|---|---|---|
| | | OTF Rescoring | Fusion $F = 50$ | OTF Rescoring | Fusion $F = 50$ |
| | $s = 0.6$ | $\delta = 0.6$ | $\delta = 0.8$ | $\delta = 0.6, \lambda = 2.0$ | $\delta = 0.9, \lambda = 1.5$ |
| Anti-Biasing, without-biasing WER: **1.7** | | | | | |
| $B = 150$ | 1.9 | 1.9 | 2.0 | 1.9 | 2.1 |
| $B = 600$ | 2.1 | 2.1 | 2.3 | 2.1 | 2.3 |
| $B = 3000$ | 2.2 | 2.2 | 2.4 | 2.2 | 2.5 |
| With-Prefix, without-biasing WER: 9.6 | | | | | |
| $B = 150$ | 1.5 | 1.0 | 0.9 | 0.9 | **0.8** |
| $B = 600$ | 1.8 | 1.3 | 1.2 | 1.0 | **0.9** |
| $B = 3000$ | 2.8 | 2.2 | 2.1 | 1.9 | **1.7** |
| Without-Prefix, without-biasing WER: 20.9 | | | | | |
| $B = 150$ | 1.8 | 0.9 | 0.8 | 1.0 | **0.8** |
| $B = 600$ | 2.1 | 1.2 | 1.0 | 1.3 | **1.0** |
| $B = 3000$ | 4.0 | 3.1 | 2.5 | 3.1 | **2.3** |
| Contact-Tag, without-biasing WER: 14.7 | | | | | |
| $B = 265$ | 3.8 | 3.5 | 3.4 | 3.1 | **3.0** |

We then perform KMP biasing on top of NAM, and the results are given in Table 2 (mid columns). We find it better to slightly tune down the biasing strength of NAM when combing it with KMP biasing, and now the optimal $\delta$ is much smaller than those used in Section 4.1, as the output of NAM already contains strong biasing information. Nonetheless, KMP provides additional 20%–40% relative WER improvements over NAM across With-Prefix and Without-Prefix, and 8%–10% relative improvements on Contact-Tag, with small degradation on Anti-Biasing.

### 4.3 BOOSTING KMP WITH PREFIXES

Finally, we verify the effectiveness of prefixes and Algorithm 5 on top of NAM + KMP biasing. We provide three prefixes {call, open, play} as additional inputs to KMP biasing while NAM only uses biasing phrases as before. For With-Prefix and Contact-Tag, each test utterance comes from one of App, Contact, and Song domains, and so it contains one of the prefix while the other two act as distractors; Without-Prefix does not contain prefixes before biasing phrases by design.

We fix NAM biasing strength to $s = 0.5$, tune the score boosting factor $\lambda$ over {1.5, 2.0, 2.5}, and search $\delta$ locally around the optimal values found in Section 4.2. Final results are shown in Table 2 (right columns). We obtain further gains on With-Prefix and Contact-Tag, while not degrading much on other sets. In particular, we achieve a final 21% relative WER reduction on Contact-Tag with shallow fusion over NAM itself. We also observe that OTF rescoring prefers a larger $\lambda$ than shallow fusion, as it has less chance to be confused by mis-recognized prefixes and wrongly boosted bonuses. It is future work to conduct full-fledged experiments with more complex prefixes.

### 5 CONCLUSIONS

We have proposed a TPU-friendly implementation of pattern-matching based biasing, and demonstrated the effectiveness of its variants on large-scale voice search queries. Our method achieves significant WER reduction on biasing sets without introducing additional learning parameters, and is complementary to a strong model-based biasing method. There are several directions for future research. To scale up our method to more than thousands of biasing phrases, we may study deep integration with NAM+ (Munkhdalai et al., 2023), which performs a quick filtering of a large amount of unlikely biasing phrases before conducting more careful search over the remaining ones. Our current implementation has used a fixed per-token score, and it is straightforward to incorporate an external neural language model for improved bonus computation (Le et al., 2021). Finally, our on-TPU implementation enables training with matching-based biasing, and it is an interesting research question to design a suitable learning objective for further performance gain.

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

## A  BUILDING THE FAILURE FUNCTION

We provide the pseudo code for building the failure function in Algorithm 4.

---

**Algorithm 4** Building failure function $\bar{\pi}$ for a single pattern.

---

**Input:** Search pattern $\mathcal{P}$ of length $m$
**Output:** Failure function $\bar{\pi}$ stored in array $\Pi$

$\quad \Pi[0] \leftarrow -1$
$\quad k \leftarrow 0$ $\hfill \triangleright$ Index of next possible match
$\quad$**for** $i = 1, \ldots, m-1$ **do**
$\quad\quad$**if** $\mathcal{P}[i] = \mathcal{P}[k]$ **then**
$\quad\quad\quad \Pi[i] \leftarrow \Pi[k]$ $\hfill \triangleright$ Shortcut
$\quad\quad$**else**
$\quad\quad\quad \Pi[i] \leftarrow k$ $\hfill \triangleright \Pi[i] \leftarrow \pi(i)$
$\quad\quad\quad$**while** $k \geq 0$ and $\mathcal{P}[i] \neq \mathcal{P}[k]$ **do**
$\quad\quad\quad\quad k \leftarrow \overline{\Pi}[k]$
$\quad\quad\quad$**end while**
$\quad\quad$**end if**
$\quad\quad k \leftarrow k + 1$ $\hfill \triangleright$ Loop invariance: $k = \pi(i+1)$ at end of each **for**-loop
$\quad$**end for**=0

---

# B  KMP BIASING WITH PREFIXES

In Algorithm 5 we provide the implementation of bonus computation with a set of prefixes. The new scoring function is defined as

$$\mu_\lambda((i^1, m^1), \ldots, (i^B, m^B)) = \max_{b=1,\ldots,B} \lambda^{\text{int}(m^i)} \cdot f(\mathcal{P}^b, i^b),$$

where $\lambda > 1$ is the score multiplier that is effective immediately after matching a prefix, and $\text{int}(m)$ converts the boolean variable $m$ into integer, i.e., $int(m) = 1$ if $m = True$ and 0 otherwise.

---

**Algorithm 5** Compute bonus score of a token extension with prefixes.

---

**Input:** Prefixes $\{\mathcal{R}^c\}_{c=1}^C$, current partial matching lengths for prefixes $\mathcal{J} = (j^1, \ldots, j^C)$. Biasing phrases $\{\mathcal{P}^b\}_{b=1}^B$, current partial matching lengths for biasing phrases. $\mathcal{I} = (i^1, \ldots, i^B)$. Current prefix matching mask $\mathcal{M} = (m^1, \ldots, m^B)$, biasing bonus multiplier $\lambda$, new token $x$.

**Output:** Updated partial matching lengths, updated prefix matching mask, and biasing bonus.

   **procedure** COMPUTEBONUS($\{\mathcal{P}^b\}_{b=1}^B, \mathcal{I}, \{\mathcal{R}^c\}_{c=1}^C, \mathcal{M}, x$)

      $prev\_score \leftarrow \mu_\lambda((i^1, m^1), \ldots, (i^B, m^B))$

      $match\_any\_bias \leftarrow False$

      **for** $b = 1, \ldots, B$ **do**

         $u^b, \ match^b \leftarrow \text{FORWARD}(\mathcal{P}^b, i^b, x)$

         **if** $match^b$ **then**

            $match\_any\_bias \leftarrow True$

            $v^b \leftarrow len(\mathcal{P}^b)$       ▷ For full match, use pattern length for computing potential

            $ext^b \leftarrow True$       ▷ $ext$ tracks if $x$ extends matching for $\mathcal{P}^b$

         **else**

            $v^b \leftarrow u^b$

            **if** $u^b > i^b$ **then**

               $ext^b \leftarrow True$

            **else**

               $ext^b \leftarrow False$

            **end if**

         **end if**

         $m^b \leftarrow (m^b \ \& \ ext^b)$       ▷ Update prefix-matching status

      **end for**

      $updated\_score \leftarrow \mu_\lambda((v^1, m^1), \ldots, (v^B, m^B))$

      $bonus \leftarrow updated\_score - prev\_score$

      $match\_any\_prefix \leftarrow False$

      **for** $c = 1, \ldots, C$ **do**       ▷ Try to consume $x$ with prefixes

         $n^c, \ match^c \leftarrow \text{FORWARD}(\mathcal{R}^c, j^c, x)$

         **if** $match^c$ **then**

            $match\_any\_prefx \leftarrow True$

         **end if**

      **end for**

      **for** $b = 1, \ldots, B$ **do**

         $restart^b \leftarrow match\_any\_prefix \ \& \ (\text{not } ext^b)$

         **if** $restart^b$ **then**       ▷ Restart matching of $\mathcal{P}^b$ after fully matching prefix

            $u^b \leftarrow 0, \ m^b \leftarrow True$       ▷ Cancel current partial matching of $\mathcal{P}^b$

         **end if**

      **end for**

      **if** $match\_any\_bias$ **then**

         $n^c \leftarrow 0 \ \ \text{for} \ \ c = 1, \ldots, C$

         $u^b \leftarrow 0, \ m^b \leftarrow False \ \ \text{for} \ \ b = 1, \ldots, B$

      **end if**

      **return** $((n^1, \ldots, n^C), \ (u^1, \ldots, u^B), \ (m^1, \ldots, m^B), \ bonus)$

   **end procedure**

---

## C WER OF KMP BIASING FOR VARYING $\delta$

We provide WER curves of varying $\delta$ in Figure 1 for both OTF rescoring and shallow fusion options.

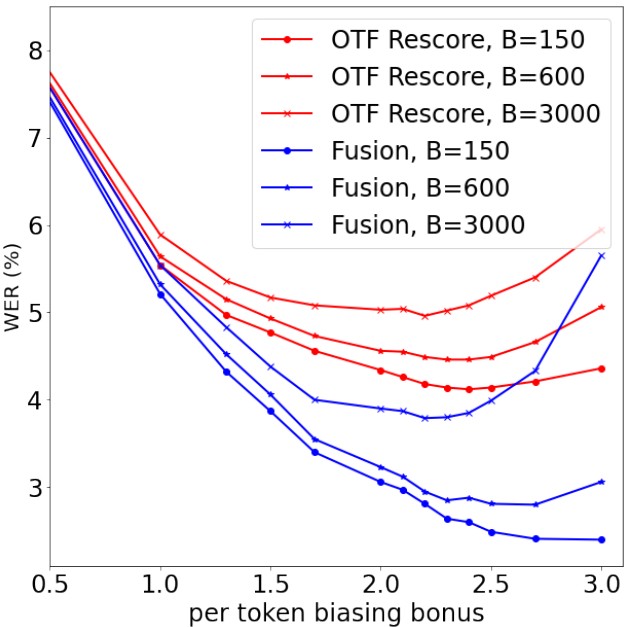

Figure 1: WER versus $\delta$ for KMP biasing in both OTF rescoring and shallow fusion ($F = 50$) modes, on the With-Prefix set.

