# OpenReview forum: "Contextual Biasing with the Knuth-Morris-Pratt Matching Algorithm"
_ICLR.cc/2024/Conference — Submitted to ICLR 2024_

### Official Review · Reviewer_ZLRq · 2023-10-30

**Soundness:** 2 fair
**Presentation:** 1 poor
**Contribution:** 2 fair
**Rating:** 3
**Confidence:** 4

**Summary:**

This paper proposes a new method for contextual biasing of speech recognition (ASR) at decode time. The usual practice is to build a WFST for the biasing phrases and use this in a kind of shallow fusion strategy during decoding. The authors posit that such a method is inefficient on TPUs, and hence propose an equivalent string matching technique based on Knuth-Morris-Pratt (KMP) algorithm. They show that this method provides significant improvements in WERs for utterances containing biasing phrases, without causing large degradations when they are not present.

**Strengths:**

1. Contextual biasing in ASR is an important application. As mentioned by the authors (Section 3), biasing can be done at the decoding level or at the model level — the proposed KMP algorithm operates at the former, showing good WER improvements. It is also shown to be complementary to model-based biasing (using NAM).

2. The authors have discussed the parallelized time and space complexities of the proposed methods wherever applicable.

3. On the Contact-Tag data, the WER is improved from 14.7% to 7.7% (for KMP) and 3.0% (for NAM + KMP), which is quite a large improvement. At the same time, the WER for “anti-biasing” only degrades from 1.7% to 2.3% and 2.5%, respectively.

**Weaknesses:**

### Motivations misaligned with application and results

The main objective of the paper is to build a contextual biasing system that is efficient to decode on large-scale parallelizable infrastructure such as TPUs. However, in the introduction and the experiments, the application of the method is for recognizing contact names for voice assistants. In my understanding, such voice assistants are commonly placed on the edge device, which does not usually have built-in TPUs. As such, it is hard to see what would be the impact of the proposed method from a decoding efficiency perspective.

Even if we ignore the above, it is hard to buy into the “efficiency” argument, since the authors do not provide any RTF results to back their claims. “Memory footprint” and “efficiency on TPUs” are essential motivating factors behind the proposed method, but the evaluation is only conducted for quality (WERs). In fact, it appears that the stated TPU-based vectorization is essentially just parallelization of a loop over all biasing phrases — it is hard to see why such parallelization would be TPU-specific.

Throughout the paper, the authors have mentioned that FST-based biasing poses challenges for efficient TPU-based implementation. Recently, FSTs have been efficiently represented and manipulated on GPUs using specialized kernels (see the GTN and k2 projects). In fact, the Aho-Korasick algorithm has recently been used for contextual biasing on GPUs and released in the “icefall” library. Why are these methods not applicable for TPUs?


### Problems with evaluation design

I am concerned about the lack of public benchmarks or baselines in the experiments. The authors use in-house voice-assistant data from a previous work (which is not publicly available AFAIK) to conduct their evaluations, and do not release code for their method. This would make it impossible to replicate or verify the reported improvements. There are also no comparisons with any other decoding-based contextual biasing methods, although the authors seem to be quite aware of their existence (Section 3). Granted, the proposed KMP algorithm should be equivalent to the WFST-based shallow biasing approach proposed earlier, but it would be good to show this as a sanity check. It would also be useful to show how the memory requirement of the WFST-based implementation versus KMP change with increase in the size of the biasing list; the latter is absolutely essential, since this is the main motivation for using this algorithm.

### Presentation

The description of the proposed method is very dense, and may benefit from some abstraction. Contextual biasing is formulated into two stages: (i) pattern matching, and (ii) boosting matched patterns. The authors should consider presenting the two parts independently (instead of the current presentation where (ii) builds on (i)). This would also be useful to think about other algorithms for (i) and (ii) without disturbing the other.

Second, the authors rely too much on Algorithm blocks to present their method (there are 5 in total including the appendices), which breaks the flow of reading and makes the paper hard to parse. It may be beneficial to release open-source code for the details of the algorithm and use more of the space to discuss the algorithms themselves, their connections with other contextual biasing methods, and their advantages/limitations.

**Questions:**

1. In Section 1 (under “Our contributions”), the authors state that their method “can be potentially useful for other sequence transduction tasks.” Can the authors describe what other tasks may benefit from sequence matching?

2. There are several linear-time algorithms for pattern matching such as Rabin-Karp, Boyer-Moore, etc. It would be useful to include a discussion about why KMP is most appropriate for the task.

---

> ### Author Response · Authors · 2023-11-21
> **Response**
>
> We will incorporate your suggestions on paper presentation in a later version, especially the connection with other contextual biasing methods, and a discussion on the currently available specialized libraries.
>
> Your other questions have been addressed in "Rebuttal to common concerns".
>
> ## **Practicality of biasing on TPU/GPU (or in general high-performance parallel computing devices)**
>
> On-TPU/GPU biasing is practical for voice assistants, because server ASR models can be run on them to enjoy high throughput via batching. And while not all on-device ASR models have access to TPU/GPUs, modern devices like pixel phones are equipped with high performance chips, and such devices will become increasingly popular and available as their cost goes down.
>
>
> ## **Other pattern matching algorithms**
>
> Thank you for pointing out the other pattern matching algorithms (Rabin-Karp and Boyer-Moore). KMP is a classical algorithm covered in standard textbooks, and has the best worst-case complexity (for a single search pattern). It is possible that other methods offer better trade-offs in certain scenarios and may be investigated in future work.

---

> > ### Comment · Reviewer_ZLRq · 2023-11-22
> >
> > Thanks for your reply. I have read this and the common response, and submitted my recommendation to the area chair.

---

### Official Review · Reviewer_8igc · 2023-10-31

**Soundness:** 2 fair
**Presentation:** 2 fair
**Contribution:** 2 fair
**Rating:** 6
**Confidence:** 5

**Summary:**

This paper is about contextual biasing for automatic speech recognition (ASR).

Contextual biasing means: Consider Google Home or Alexa, where it is common to play songs, or maybe call someone from the contact list. So when the user speaks, the probability for such words of recently played songs or from person names from the contact list are higher than for other users, and contextual biasing will use that knowledge and boost the scores in the beam search recognition for such words or phrases.

Contextual biasing is not new, and many previous solutions to this exist. In this paper, a new method is proposed, to improve the beam search specifically. The ASR model is not changed. More specifically, the authors propose to use the Knuth-Morris-Pratt algorithm as an efficient way to find biasing phrases in the hypotheses and then boost them.

The Knuth-Morris-Pratt (KMP) algorithm is an algorithm to search for a substring in a given string in an efficient manner. The naive implementation would take O(n * m), n being the long string length, m being the substring length, while KMP can do it in O(n + m).

This provides an alternative formulation to the weighted finite state transducer framework, which conceptually does the same. However, they also use an efficient TPU-friendly implementation of this specific algorithm.

The experiments show that the KMP method on its own performs slightly worse than another model-based biasing method, namely the neural associative memory (NAM). However, KMP and NAM combined give improvements over NAM alone.

**Strengths:**

They show how to apply the KMP algorithm inside beam search to boost the biasing phrases.

The experiments show that the proposed KMP-based algorithm gives nice improvements when used together with NAM in the setting biasing.

**Weaknesses:**

The topic is very ASR specific. I'm not sure if the broader ICLR community is interested in this, and some conference like Interspeech or ICASSP would be a better fit?

In principle, the method could be applied for other tasks, for example for machine translation. However, this is not investigated here. I think this would make it a better fit for ICLR.

It is explained that the proposed method is conceptually similar (or the same?) as WFST-based approaches. However, in the experiments, it is not compared.

The argument is about KPM being TPU friendly. However, if WFST is conceptually equivalent, how can it be different? It's just a matter of implementation then. This is either stated confusingly, that it is in fact different. Or it is stated confusingly that they are the same, and you can also use the WFST formulation to describe exactly the same algorithm. In any case, it's a bit confusing. So then using the WFST-based formulation, you could just use the equivalent algorithm, and it would also be TPU friendly. In any case, this should be clarified.

The argument about being TPU friendly again: Actual speed performance is not really compared. How much worse does the WFST-based approach perform?

Code is not released?

**Questions:**

Abstract:

> Our method simulates the classical approaches often implemented in the weighted finite state transducer (WFST) framework, but avoids the FST lan- guage altogether, ...

I'm not sure what this means. Does this mean, it is actually equivalent to what is being done with the WFST framework, and only a reformulation/reinterpretation? So the novelty here is no new method, but just a new interpretation of the existing algorithm? Or is this really different? Why is it relevant to avoid the FST language? What is the actual difference when you don't avoid the FST language, i.e. when you look away from just rephrasing things.

Also, why would you want to avoid the FST language? The FST language is very simple, while the presented algorithms actually look more complicated? Maybe it would actually be helpful to not avoid the FST language, but to present the proposed method within the FST language, so that it is easier to see the actual differences, and also easier to understand.


Section 3 (but same thing also said in intro and elsewhere):

> While they [FSTs] can be represented as graph with sparse adjacency matrices, in general FSTs are not efficient to use on TPUs which are optimized for dense operations. Our work is one step towards incorporating FST functionalities into a TPU-friendly implementation.

I'm not sure if this is really about FST or not. Couldn't you reformulate the presented work also as an FST, and then this statement would be false?


I'm not sure if the paper is a bit too ASR specific? What about applying this also to machine translation or other tasks?


Runtime differences between OTF Rescoring, Fusion F = 10, Fusion F = 50, Fusion F = 4096? And the same also with NAM?


Is the code released? If not, this would be a major weakness of the work.


With increased B (number of biasing phrases), the result gets worse. First to clarify: The test dataset is designed such that the word from reference transcription is always in the biasing phrases? Or otherwise, why would it get worse with increased B? And then: in practice, how big would B be? And how to update the set of biasing phrases? Wouldn't a test make more sense which is more close to how this is actually used in production?

On TPU, what is actually parallelized? I assume, in Algorithm 2, the loop over B is parallelized? Ok, yes, you write that in the text (end of Section 2.2). It would be nice to specifically mark this in the algorithms, e.g. write "vectorized-for" or so. E.g. also in Algorithm 3, the loop over the hypotheses and also over the k=1...F would also not really be a loop but run in parallel (I assume).

Comparison to WFST-based biasing?

---

> ### Author Response · Authors · 2023-11-21
> **Reply**
>
> We thank the reviewer for constructive feedback. Your concerns are shared by other reviewers and addressed in "Rebuttal to common concerns".

---

> ### Author Response · Authors · 2023-11-22
> **Number of biasing phrases**
>
> (Just realized that we have not addressed your question on B.)
>
> The test dataset is designed such that the word from reference transcription is in the biasing phrases, except for *anti-biasing* where all provided biasing phrases are distractors. In general, we want to improve WER when ground truth is provided, and not to degrade WER when it is not. The larger the B, the worse the WER is because there are more possibilities to bias towards, and therefore more chances to confuse the ASR model.
>
> The actual number of B depends on the specific application scenario. In the case of on-device call contact biasing, the number of biasing phrases is the number of contacts in phone book, which is typically few hundreds (as in our Contact-Tag test set). In the case of biasing for geographical locations, the number of biasing phrases could be quite a few thousands or more.

---

### Official Review · Reviewer_kxsf · 2023-11-01

**Soundness:** 3 good
**Presentation:** 3 good
**Contribution:** 3 good
**Rating:** 5
**Confidence:** 4

**Summary:**

This paper addresses the problem of contextual biasing for speech recognition. The paper proposes to use very popular and efficient KMP algorithm used for pattern searching to bias the ASR decodes with the contextual terms.

Experimental results on the enterprise model and real/TTS audio show usefulness of this approach.

**Strengths:**

The paper clearly demonstrates the applicability of KMP's efficiency on the biasing task.

The paper reads well and is easy to follow.

**Weaknesses:**

1. The paper lacks adequate references to the related work in the area of the contextual biasing for ASR Models. I have added some relevant citations.

2. Lack of comparison on publicly available data and models limiting reproducibility. Le at al 2021a [2] provides an open protocol for evaluating on librispeech corpus (https://github.com/facebookresearch/fbai-speech/tree/main/is21_deep_bias).

3. Lack of comparison with baselines, how well does this model compare against a simple shallow fusion with an external n-gram trained on the dictionary items for the cases with prefixes? Comparison against the Neural baselines such as NAM and other relevant works from the literature are missing.

References:

[1] Gourav et al. 2021. Personalization strategies for end-to-end speech recognition systems. ICASSP 2021

[2] Le et al. 2021a. Contextualized streaming end-to-end speech recognition with trie-based deep biasing and shallow fusion.

[3] Le et al. 2021b. Deep shallow fusion for rnn-t personalization. IEEE Spoken Language Technology Workshop (SLT),

[4] Guangzhi Sun, Chao Zhang, and Philip C Woodland. 2021. Tree-constrained pointer generator for end-to-end contextual speech recognition.

[5] Huber et al. 2021. Instant one-shot word learning for context-specific neural sequence-to-sequence speech recognition.

[6] Guangzhi Sun, Chao Zhang, and Phil Woodland. 2023a. Graph neural networks for contextual asr with the tree-constrained pointer generator.

[7] Naowarat, Burin, et al. 2023, Effective training of attention-based contextual biasing adapters with synthetic audio for personalised ASR.

**Questions:**

1. In this work, the bonus score is added each time the match happens, how do you address the scenario where prefix has matched but suffix won't match and the hypothesis are still carrying extra weight provided during the biasing?
E.g, if the dictionary item is TWIN and the actual audio has TWENTE, but the hypothesis is preferring TWIN (and/or its continuations) due to additional biasing.

2. What are the RTF scores for this method?

---

> ### Author Response · Authors · 2023-11-21
> **Response**
>
> We thank the reviewer for pointing out the references and we will incorporate them into a later version. Your other questions have been addressed in "Rebuttal to common concerns".
>
> ## **Regarding your question 1**
>
> Assume the dictionary contains only one item “twin”. And assume that the model uses beam_size=1, and has already output “_tw”, and thus a bonus for “_tw” was added to the hyp.
>
> In the next step, the ASR model considers two possible expansions “en” and “in”.
>
> - “in” leads to longer matching of the dictionary item “twin”, so the proposal “in” receives a bonus.
>
> - “en” leads to non-matching of any dictionary item, and therefore it also has to cancel the bonus received by “_tw”. The canceling part is implied by our potential function, dependent on the longest partial matching length (which is 0 after accepting "en"), and the bonus being the difference between potentials before and after accepting the token; see Section 2.2.
>
> Therefore it boils down to which is higher, score(“in”) + bonus(“in”) versus score(“en”) - bonus(“_tw”), and the token with higher final score will be accepted and the other is pruned.
>
> It may be the case that the ASR score for “en” is much higher than that of “in”, and “twen” becomes the top hypothesis, and search continues with a total biasing bonus of 0 (due to the canceling of bonus).

---

### Author Response · Authors · 2023-11-21
**Rebuttal to common concerns (Part I)**

We thank reviewers for constructive feedback. Here we address a few common concerns (remaining questions from each individual reviewer are addressed below the original review).


## **Wide applicability of our method**

Our method is not limited to ASR. It can be used with any autoregressive sequence model, during decoding or sampling/generation, to promote desired phrases or patterns (one could potentially incorporate more regex capability such as wildcard). For example, one may use our algorithm in LLM generation, to promote certain topics/facts, by providing the model a list of relevant phrases with positive bonuses. Another usage of our method is negative biasing, i.e., if we would like to prevent certain offensive results being decoded, we could provide the list of forbidden phrases and associate them with negative bonuses (or rather, penalties).

## **Reproducibility**

We will open source our implementation, after separating KMP-biasing from the production ASR recipe. Note we have provided accurate details in the pseudo-algorithms; actual implementations are relatively straightforward to obtain, by translating the pseudo-algorithms into tensorflow with proper vectorization. We also discussed memory cost, vectorization over (hyp, extension, phrase), and fast gathering by einsum throughout Section 2.

## **Comparison on public dataset**
We first remark that our voice-search setup is realistic and challenging, which has a vocabulary of a few millions words, with long-tail distribution over a large amount of rare words. Our test sets are used in multiple prior publications, and our NAM baseline matches the most recent publication by Wu et al, 2023 (Dual-Mode NAM).

During the rebuttal period, we manage to experiment with public Librispeech setup by:

 - Le et al. 2021a. Contextualized streaming end-to-end speech recognition with trie-based deep biasing and shallow fusion.

Note Librispeech has a vocabulary of 200K words, and all but the top-5K most frequent words are considered rare in this work. Also the OOV rate for Librispeech testsets are low, at 0.6% and 0.8% for test-clean and test-other respectively, see <https://arxiv.org/pdf/1902.01955.pdf> (Table 1); these are the reasons why we think "non-public" voice-search setup is more important, even though it's proprietary.

We have trained a 300M CTC model, which is a weaker acoustic model than RNN-T used by Le et al 2021a, and performed NAM and KMP biasing on top of CTC. Nonetheless, we obtain improved WERs with both KMP-biasing and NAM+KMP, compared with the “s3” setup, i.e., Deep Biasing-RNNT + WFST, which is the strongest model without external neural LM in Le et al. 2021a. Results for different number of biasing phrases are shown in the table below.

|                        | B=100 |  | B=500 | | B=1000 | | B=2000 ||
| ---------------------- | -----: | -----: | ------: | ------: | -----: | -----: | ------: | ------: |
| Model                  | clean | other | clean  | other  | clean | other | clean | other |
| Le et al, RNN-T        | 3.65  | 9.61  | 3.65   | 9.61   | 3.65 | 9.61 | 3.65 | 9.61 |
| Le et al, DB-RNNT+WFST | 2.81  | 8.10   | 2.91   | 8.30    | 3.00 | 8.45 | 3.04 | 8.75 |
| Ours, CTC              | 4.17  | 10.22 | 4.17   | 10.22  | 4.17 | 10.22 | 4.17 | 10.22 |
| Ours, NAM                    | 3.31  | 8.72  | 3.44   | 8.96   | 3.48 | 9.09 | 3.57 | 9.28 |
| Ours, KMP                    | **2.65**  | **7.45**  |  **2.80**    |  **7.81**   | **2.88** | **8.09** | 3.07 | **8.45** |
| Ours, NAM+KMP                | **2.33**  | **6.84**  | **2.52**   |  **7.26**   | **2.65** | **7.53** | **2.86** | **7.97** |

We expect that using our methods with a stronger RNN-T system will further improve WERs.

---

### Author Response · Authors · 2023-11-21
**Rebuttal to common concerns (Part II)**

## **Runtime benchmark**

Our model is non-streaming (i.e., processes the entire audio before producing results) and the encoder output has a frame rate of 320ms due to time reduction. We measure the run time of our ASR system on a single Google cloud TPU v3 (https://cloud.google.com/tpu/docs/system-architecture-tpu-vm#tpu_v3), for processing a batch of 8 utterances of 15-long random audio, with both B=300 biasing phrases and B=1000 biasing phrases.

| Component                             | Runtime for B=300 | Runtime for B=1000 |
|---------------------------------------|-------------------: |--------------------: |
| Encoder for 15s audio                 | 65 ms              | 65 ms               |
| NAM                                   | 10.4 ms            | 34.6 ms             |
| Per Beam search step without biasing  | 0.38 ms            | 0.38 ms             |
| Per Beam search step KMP, fusion F=50 | 0.79 ms            | 1.5 ms              |
| Per beam search step KMP, fusion F=10 | 0.25 ms            | 0.4 ms              |
| Per beam search step KMP, OTF (F=1)   | 0.1 ms             | 0.21 ms             |

Cost of preparation work (biasing phrase encoding for NAM, and failure function computation for KMP) are not shown as they can be done in parallel to audio encoder (and take less time). For much larger # of phrases, one shall perform a quick filtering of biasing phrases (e.g., based on NAM), before sending them for decoding.

Also note that TPU v3 was launched around 2018, and more recent TPU versions could be a few times faster.

## **CPU WFST-biasing baseline**

It is highly non-trivial to set up CPU WFST biasing for the exact model we have in the submission and finish tuning in time. Here we provide biasing results for a model similar to the “large-pass” model from the following paper:

- Ding et al. A Unified Cascade Encoder ASR Model for Dynamic Model Sizes. 2022.

This model has 120M parameters, was trained on 1 million hours (more than the amount of data used in current paper), with a 4.2% WER on the in-house voice search eval set (on which the 900M model in current paper achieves 3.8%).

This model achieves 14.7% WER on Contact-Tag without biasing, matching that of the model in the current paper. WFST-biasing with this model with extensive list of carrier phrases (e.g., "call \\$CONTACT", "can you connect me to \\$CONTACT", "please dial \\$CONTACT") achieves 3.9%, which is close to the WER of NAM and better than WER of KMP without carrier phrases. These results provide a good sanity check. We are in the process of benchmarking the size and runtime of FST and will provide the results when they are available.

## **Significance of our approach and “avoiding the FST language”**

We shall clarify what we meant by “avoiding the FST language” in a later version. The graph representations of FSTs are generally sparse (there exist no edges between most node pairs). An alternative to our approach is to represent the FST as a list of edges, of the form (start state, end state, input/output label, weight), and perform lookup of these edges to follow the FST on TPU/GPU. Note that this approach still requires first constructing the FST with non-trivial optimizations (e.g., determinization, minimization, weight pushing, etc). Our approach has a clear advantage in its simplicity: as a reviewer commented, it boils down to simply translating our pseudo-algorithms into any deep learning platform with a few loops, without any special FST operations (although realizing its equivalence to WFST requires understanding of automata theory).

Our on-TPU benchmarks show that our method is efficient. The small memory footprint is self-evident: for each utterance we store the failure function table which has the same size of the biasing phrases, and for each hypothesis we store an integer (partial matching length) for each phrase. We did perform KMP biasing with 20K phrases, and the method fits on TPU v3 equipped with 16G memory.

Another important advantage of our method is that it trivially allows automatic differentiation throughout the decoding process (implemented with tf.while loop), and opens the door to learning with the discrete structure of biasing (e.g., by discriminative training).

---

### Meta-Review · Area_Chair_iJju · 2023-12-06

**Metareview:**

The paper proposes an application of the Knuth-Morris-Pratt (KMP) pattern matching algorithm as an extension of beam search for ASR decoding. This aims to perform "contextual biasing" to boost the chance that the output will contain rare but contextually-relevant entities. The approach is shown to reduce WER on an English voice search query benchmark. The main weaknesses are that the paper's presentation and use of terminology is not clear enough, that the experimental results lack some relevant baselines, and that the results do not sufficiently support the claims about efficiency. It is also unclear if the method is more generally applicable outside the specific ASR application.

**Justification For Why Not Higher Score:**

The proposed method is very ASR-specific, and there are a number of issues with the experimental results and clarity of the paper.

**Justification For Why Not Lower Score:**

N/A

---

### Decision · Program_Chairs · 2024-01-16

Reject